# The “Historical Materials BAG”: A New Facilitated Access to Synchrotron X-ray Diffraction Analyses for Cultural Heritage Materials at the European Synchrotron Radiation Facility

**DOI:** 10.3390/molecules27061997

**Published:** 2022-03-20

**Authors:** Marine Cotte, Victor Gonzalez, Frederik Vanmeert, Letizia Monico, Catherine Dejoie, Manfred Burghammer, Loïc Huder, Wout de Nolf, Stuart Fisher, Ida Fazlic, Christelle Chauffeton, Gilles Wallez, Núria Jiménez, Francesc Albert-Tortosa, Nati Salvadó, Elena Possenti, Chiara Colombo, Marta Ghirardello, Daniela Comelli, Ermanno Avranovich Clerici, Riccardo Vivani, Aldo Romani, Claudio Costantino, Koen Janssens, Yoko Taniguchi, Joanne McCarthy, Harald Reichert, Jean Susini

**Affiliations:** 1European Synchrotron Radiation Facility, 71 Avenue des Martyrs, 38000 Grenoble, France; catherine.dejoie@esrf.fr (C.D.); burgham@esrf.fr (M.B.); loic.huder@esrf.fr (L.H.); wout.de_nolf@esrf.fr (W.d.N.); stuart.fisher@esrf.fr (S.F.); ida.fazlic@esrf.fr (I.F.); mccarthy@esrf.fr (J.M.); reichert@esrf.fr (H.R.); jean.susini@synchrotron-soleil.fr (J.S.); 2Laboratoire d’Archéologie Moléculaire et Structurale (LAMS) CNRS UMR 8220, UPMC Univ Paris 06, Sorbonne Université, 5 place Jussieu, 75005 Paris, France; 3Université Paris-Saclay, ENS Paris-Saclay, CNRS, PPSM, 91190 Gif-sur-Yvette, France; 4Antwerp X-ray Imaging and Spectroscopy laboratory (AXIS) Research Group, NANOLab Centre of Excellence, University of Antwerp, Groenenborgerlaan 171, 2020 Antwerp, Belgium; ermanno.avranovichclerici@uantwerpen.be (E.A.C.); koen.janssens@uantwerpen.be (K.J.); 5Paintings Laboratory, Royal Institute for Cultural Heritage (KIK-IRPA), Jubelpark 1, 1000 Brussels, Belgium; 6CNR-SCITEC, c/o Department of Chemistry, Biology and Biotechnology, University of Perugia, Via Elce di Sotto 8, 06123 Perugia, Italy; aldo.romani@unipg.it (A.R.); claudio.costantino@studenti.unipg.it (C.C.); 7Centre of Excellence SMAArt and Department of Chemistry, Biology and Biotechnology, University of Perugia, Via Elce di Sotto 8, 06123 Perugia, Italy; 8Rijksmuseum, Conservation and Restoration, P.O. Box 74888, 1070 DN Amsterdam, The Netherlands; 9Chimie ParisTech, PSL University, CNRS, Institut de Recherche de Chimie Paris, 11 rue Pierre et Marie Curie, 75005 Paris, France; c.chauffeton@chimieparistech.psl.eu (C.C.); gilles.wallez@sorbonne-universite.fr (G.W.); 10Cité de la Céramique Sèvres-Limoges, place de la Manufacture, 92310 Sèvres, France; 11Centre de Recherche et Restauration des Musées de France (C2RMF), Porte des Lions, 14 quai François Mitterrand, 75001 Paris, France; 12UFR 926, Sorbonne Université, 75005 Paris, France; 13Departament d’Enginyeria Química EPSEVG, Universitat Politècnica de Catalunya (UPC)·BarcelonaTech Av. Víctor Balaguer s/n, 08800 Vilanova i la Geltrú, Spain; nuria.jimenez.garcia@upc.edu (N.J.); francesc.albert.tortosa@upc.edu (F.A.-T.); nativitat.salvado@upc.edu (N.S.); 14Institute of Heritage Science, National Research Council, ISPC-CNR, Via R. Cozzi 53, 20125 Milan, Italy; elena.possenti@cnr.it (E.P.); chiara.colombo@cnr.it (C.C.); 15Politecnico di Milano, Physics Department, Piazza Leonardo da Vinci 32, 20133 Milano, Italy; marta.ghirardello@polimi.it (M.G.); daniela.comelli@polimi.it (D.C.); 16Department of Materials Science and Engineering, 3mE, Delft University of Technology, Mekelweg 2, 2628 CD Delft, The Netherlands; 17Pharmaceutical Science Department, University of Perugia, Via del Liceo 1, 06123 Perugia, Italy; riccardo.vivani@unipg.it; 18History and Anthropology, Faculty of Humanities and Social Sciences, University of Tsukuba, 1-1-1 Tennodai, Tsukuba 305-8577, Japan; taniguchi.yoko.fu@u.tsukuba.ac.jp

**Keywords:** synchrotron, X-ray diffraction, cultural heritage, beamtime access, paintings, pigments, ceramics, artistic, crystallography, structural analyses

## Abstract

The European Synchrotron Radiation Facility (ESRF) has recently commissioned the new Extremely Brilliant Source (EBS). The gain in brightness as well as the continuous development of beamline instruments boosts the beamline performances, in particular in terms of accelerated data acquisition. This has motivated the development of new access modes as an alternative to standard proposals for access to beamtime, in particular via the “block allocation group” (BAG) mode. Here, we present the recently implemented “historical materials BAG”: a community proposal giving to 10 European institutes the opportunity for guaranteed beamtime at two X-ray powder diffraction (XRPD) beamlines—ID13, for 2D high lateral resolution XRPD mapping, and ID22 for high angular resolution XRPD bulk analyses—with a particular focus on applications to cultural heritage. The capabilities offered by these instruments, the specific hardware and software developments to facilitate and speed-up data acquisition and data processing are detailed, and the first results from this new access are illustrated with recent applications to pigments, paintings, ceramics and wood.

## 1. Introduction

Synchrotron radiation facilities are increasingly used to study ancient materials from cultural heritage [1,2,3]. Assets of synchrotron radiation (SR)-based techniques are many, in particular including the following: (i) the beam brightness, which offers small probes (down to tens of nanometers) combined with high beam intensity (>10^12^ photons per seconds) and consequently high acquisition speed and high data quality; (ii) the energy tunability, a key property for absorption spectroscopy techniques, from X-ray to UV-vis and infrared range. Many techniques such as X-ray fluorescence (XRF), X-ray powder diffraction (XRPD), X-ray absorption spectroscopy (XAS), and phase contrast tomography are now commonly used for the study of art history and for the knowledge and conservation of our cultural and natural heritage [2]. More specifically, the possibility to use these methods with mapping capabilities, sub-micrometric resolution and high sensitivity makes them highly suitable for the multi-modal micro-analysis of highly heterogeneous and multi-layered tiny fragments from artifacts [1].

The main assets of XRPD-based techniques, as compared to the other techniques listed above, is not only the possibility to identify crystalline phases within complex mixtures, but also to obtain information regarding their crystallite size, orientation, microstrain and their crystal structure. Several XRPD-based set-ups and beamlines are available and complement each other. They differ in their technical characteristics, namely maximum sample size and weight, sample environment, beam size range, X-ray energy range, X-ray spectral bandwidth, detection modalities, acquisition speed, level of automation, versatility, etc. Notably, two complementary XRPD-based techniques are increasingly used by the cultural heritage community: high angular resolution X-ray powder diffraction (HR-XRPD) and micro X-ray powder diffraction (µXRPD) mapping. The former is key for the accurate characterization of crystalline materials (e.g., identification and quantification, structure refinement and crystallite size determination, description of the crystallographic structure). The latter yields more qualitative information but provides additional insight into the 2D or 3D distribution of crystalline phases at the micrometre scale. When applied to historical materials, these analyses can give clues about both the early life of the materials (creation) and later phases of the life of the materials (degradation, past and present conservation interventions) [4,5]. The same pigment can exist in different compositions, as a consequence of different synthesis procedures (by varying the nature and ratio of starting reagents, pH, etc.) or refinement steps. This can translate into different pigment grades, colors, prices, stability, etc. [4,6]. Identifying precisely such formulations can give insight into the decision of an artist to choose one or another quality of the same pigment or its availability in a specific time period or region. As an example, lead white, a ubiquitous pigment used since antiquity for paintings, is usually composed of a mixture of cerussite (PbCO_3_) and hydrocerussite (Pb_3_(CO_3_)_2_(OH)_2_). Different ratios of these compounds and different crystallite sizes can be associated with different post-synthesis treatments of the white powder obtained after lead corrosion following the traditional production of lead white [7]. This information was used to explain the presence of different lead whites in Leonardo da Vinci’s *Virgin and Child with St. Anne* [8]. Determining precisely the pigment composition is also very important for conservation purposes. As an example, the various chrome yellows (PbCr_1−x_S_x_O_4_, with 0 ≤ x ≤ 0.8) used by Van Gogh in *the Sunflowers* have different yellow-orange hues, related to their different chemical composition and crystalline structure, each with their own photochemical stability [6,9,10,11]). In general, in addition to crystalline phase identification, determining their distribution within a multi-layer system can reveal the way the artist applied artistic materials (e.g., in a painting process), but it can also highlight materials which were not in the original artwork, for example materials added during later modifications of the artwork (by the artist, by conservators), or materials formed or deposited during degradation processes.

At synchrotron radiation facilities, continuous efforts are dedicated not only to optimize techniques in terms of speed, lateral resolution, sensitivity, but also to implement new techniques. This usually relies on the implementation of new optics, mechanics, electronics, detectors, and less often on the upgrade of the synchrotron source itself. The European Synchrotron Radiation Facility (ESRF, Grenoble, France) has been benefitting from a major upgrade program (EBS—Extremely Brilliant Source), its main component being a revolutionary new electron storage ring concept that increases the brilliance and coherence of the X-ray beams produced by a factor of 100 (brilliance up to some 10^22^ ph/s/0.1%/mm^2^/mrad^2^). The full exploitation of the ESRF-EBS calls for new paradigms in order to address new long-term sustainability challenges related to the unprecedented X-ray beam properties. By way of example, some experiments which would have typically required several hours per sample can be carried out now in a few minutes. However, the time spent in writing a two-page proposal, completing its technical and scientific review, in planning the experiment, in setting-up the beamline, etc., stays fundamentally the same. It was therefore essential to develop new access routes to ESRF beamlines. In the context of the H2020 European project STREAMLINE, new access modes are in development at the ESRF, inspired by the success of the block allocation group (BAG) system to schedule beamtime used in structural biology for several years (https://www.esrf.fr/CommunityAccess (accessed on 1 February 2022)). This entails grouping together user experiments requiring the same beamline set-up in a single project rather than having many users submitting individual projects for one or two shifts (e.g., 8 h of beamtime). This not only saves time in assessing proposals and setting-up the beamlines, but also maintains users’ flexibility in the choice of projects and samples, and fosters collaborations and synergy within user communities. Ten European institutes (Rijksmuseum, Amsterdam, The Netherlands; TU Delft, The Netherlands; CNR-SCITEC, Perugia; Courtauld Institute of Art, London, UK; Politecnico di Milano; Centre de Recherche et de Restauration des Musées de France, Paris, France; Institut de Recherche de Chimie de Paris, Paris, France; Universitat Politècnica de Catalunya, Barcelona, Spain; University of Antwerp, Belgium and the ESRF, Grenoble, France) have proposed such a (Heritage) BAG for structural investigations of historical materials. Within the Historical Materials BAG, different projects are grouped together that all require structural information obtainable by X-ray powder diffraction at the ESRF, either through µXRPD/µXRF mapping at ID13 or HR-XRPD at ID22. Through the Heritage BAG, regular access to ID13 and ID22 (once every six months) is provided for a 2-year period (2021–2023) to the partners, which is renewable upon request and reviewed at the end of the 2-year period.

Below we present the experimental set-ups offered through the BAG, the hardware and software developments implemented in the context of the BAG, and some recent applications. Most of the applications are related to pigments and paintings; however, the instruments can be exploited to analyze any artistic materials, as illustrated with the examples of enamels and wood.

## 2. High-Lateral Resolution 2D X-ray Diffraction Mapping at ID13

### 2.1. Beamline Description

ID13 is an ESRF undulator beamline dedicated to high-lateral-resolution diffraction and scattering experiments using focused monochromatic X-ray beams [12]. Two end-stations, a micro-branch (beam size ~2 × 2 µm^2^) and a nano-branch (beam size down to 100 nm), are operated in time-sharing mode. For the BAG, the microbranch was preferred for different reasons: (i) a larger beam, to better fulfil powder diffraction conditions over single-crystal-like diffraction when the beam size is smaller than the crystallite size; (ii) a large range of sample stage scanning motors, allowing to mount large samples (used for example to analyze centimetric papyrus fragments) or large sample holders (see next section). Samples are mounted vertically, perpendicular to the X-ray beam. The energy of the incident beam is chosen around 13.0 keV, typically in the pre-edge region of Pb, a ubiquitous element in paintings. The energy is usually chosen to slightly excite Pb L_3_-edge XRF, but without saturating the XRF detector, nor attenuating too much the transmission of the beam through the material, and consequently the XRPD intensity. The beam is focused to ~2 × 2 µm^2^ (flux ~2 × 10^12^ ph/s, at I = 128 mA electron beam current) using a compound refractive lens set-up (CRL) mounted in a transfocator. For the study of beam-sensitive samples, the flux can be reduced (typically by a factor of 10) by detuning the gap of the undulator. XRPD maps are obtained by raster-scanning the samples and collecting 2D XRPD patterns, in transmission, with a Dectris EIGER 4 M single photon counting detector that acquires frames with 2070 × 2167 pixels (75 × 75 μm^2^ pixel size) at a rate up to 750 Hz. A dwell time of 10 ms is usually sufficient to detect most of the crystalline phases, with a reduced risk of beam damage (see below). XRF spectra are collected simultaneously with XRPD patterns, using a Vortex EM detector and XIA readout electronics. A detailed description of the set-up is shown in Appendix A.

### 2.2. Sample Preparation and Mounting

For optical and electron microscopic observations, historical materials are regularly prepared as transversal cross-sections. Such preparations are compatible with the ID13 set-up if the sample dimension, composition and density in the direction perpendicular to the cross-section surface allow for sufficient transmission of X-rays at ~13 keV. The resin blocks should be resized to the minimum dimensions in three directions, in order to reduce the X-ray absorption by the resin and the space necessary to mount each sample on the sample holders. Users are strongly encouraged, whenever possible, to prepare thin sections, which offer a much better control of sample thickness and consequently of X-ray transmission. Using thin sections allows having a controlled and homogenous probed voxel over the 2D surface [13]. For this purpose, the ID21 microtome is regularly used to prepare thin sections of ~5–10 µm. In the case of paint mock-ups applied on polycarbonate sheets, sectioning can be performed without any prior embedding of the samples [14]. If such sections are sufficiently large (>~1 mm) or in the case of a slice from a resin-embedded sample, slices can be glued on two edges, keeping the 2D analysis region completely free of any mounting material (Appendix A, right). Alternatively, and more particularly in the case of historical small and precious samples, and/or when the fragments are fragile and prompt to break when sliced, a piece of tape can be deposited on the surface of the cross-section to maintain the paint structure during sectioning. This procedure as well as the preparation of thin sections from existing and precious historical cross-sections are described in detail in the supporting information of [15].

For materials which cannot be sliced with a microtome (e.g., glass, ceramics), a procedure consisting of double-side polishing is recommended (for further details see supporting information of [16]).

Cross-sections and thin sections are then mounted on specific sample holders (Appendix A). This step is crucial to the success of the experiment. Indeed, because of the design of the ID13 microscope, which is equipped downstream with an on-axis optical microscope, changing the sample set requires a long (~30 min) procedure (see details in Appendix A). The success of the BAG relies upon the user mounting as many samples as possible on the same sample holder to reduce the set-up time with respect to data acquisition time. To benefit from the higher speed of the horizontally scanning motor, the direction where the sample is most heterogeneous is usually oriented horizontally.

### 2.3. Data Acquisition

While the ID13 set-up is controlled by BLISS commands, all the steps related to samples (navigation, focus and selection of regions and points of interest (ROIs and POIs), data acquisition) can be performed easily by non-expert users thanks to the graphical user interface (GUI) Daiquiri (cf. Appendix A). It was primarily developed for ID21 [17] and was recently deployed to ID13, and successfully commissioned and used for the BAG project. This GUI is another key element for the success of the BAG. In only a few minutes, any untrained user can perform the following:(1)Define the sample name (which will automatically define the structure of data saving, with one folder per sample), and possibly add comments about the sample;(2)Navigate on the sample holder by clicking on plus/minus steps on the sample stage motors or by clicking directly on the video image. A mosaic photograph of the entire sample holder (collection of optical images taken while raster-scanning the sample holder over 2D large regions) permits the user to observe, grab and queue positions of interest for all the samples at once;(3)Define a ROI over the 2D area to be scanned. A unique number is associated with each ROI, which will be used in data naming;(4)Select the conditions for each map (pixel size, dwell time, detector(s), low/high flux (LF/HF) to mitigate beam damage, see below), position of the XRF detector (to mitigate detector saturation);(5)As an alternative to standard XRPD mapping, a single-point acquisition mode has been implemented for the purpose of beam damage studies. This mode allows the selection of POIs and the repeated acquisition of thousands of XRPD patterns at unique positions, to monitor the evolution in the XRPD patterns (peak position, intensity and width) as a function of accumulated dose;(6)Build a queue of all the above ROIs and POIs scans and organize them along a priority list.(7)Once the experimental set-up is back to data acquisition mode (see Appendix A), the queue can be easily launched and continuously indicates the on-going and remaining scans.(8)Data produced during each scan is given a unique and automatic identifier and a proper place within the experiment folder.(9)Daiquiri also offers the possibility to visualize results in real time, such as XRF emission or XRPD intensity over a pre-set range of channels or angles, respectively. These images can be displayed, superimposed on the registered optical light image, providing in real time first diagnostics about the sample composition.

The so-called ICAT tools implemented for the ESRF data policy strategy (https://www.esrf.fr/datapolicy (accessed on 1 February 2022)) offer four panels: (i) the dataset list (with metadata of each sample and tools for downloading data); (ii) the electronic logbook (automatically filled with BLISS/Daiquiri command lines and error messages, but also appended by users); (iii) shipping information for remote experiments; (iv) information on and management of experiment participants.

Last but not least, the remote access mode, Guacamole, implemented at the ESRF as a remedy to the limited access to the facility imposed by COVID-19, makes all these steps easily available to anyone worldwide. This is an important step for the cultural heritage community, giving a chance to non-expert end-users (conservators, art historians, archaeologists, etc.) to join remotely the on-going experiments and use the X-ray beamline as easily as they would use an optical microscope at home.

### 2.4. Data Processing and Data Analysis

Data are produced as .h5 files, following the NeXuS convention and the ESRF data policy. The 2D XRPD patterns are azimuthally integrated using dedicated Jupyter notebooks, based on the PyFAI software package [18]. The processing notebooks are open-source and freely available at https://gitlab.esrf.fr/loic.huder/juno (accessed on 1 February 2022). The first two notebooks (0 and 1) are used for the calibration of the set-up. The next two (2a and 2b) deal with the azimuthal integration of 2D maps and series of repeated scans, respectively. After having defined the sample and dataset names as well as the integration parameters, any non-expert user can run the notebook through the Jupyter interface. The plots of the average integrated XRPD pattern, and the map (or series) of the integrated XRPD intensity shown in the notebooks offer a primary diagnostic of data quality and processing. Integrated data is saved as .h5 files and can be converted to .edf format by the notebook.

For the analysis of XRPD maps, the XRDUA software offers the most advanced tools for a precise, quantitative Rietveld refinement-based fit of data in .edf format [19]. Alternatively, the PyMca ROI imaging software [20] offers tools for the calculation of map intensities over 2-theta/q regions of interest, for the creation of average XRPD patterns over a selection of pixels, or for performing principal component analyses, non-negative matrix approximation calculations, etc. PyMca also offers the possibility to batch-fit XRF data, as well as the combined analysis of XRPD and XRF data in master/slave panels. Finally, a third Jupyter notebook offers tools for fast fitting of the .h5 files, as a linear combination of a set of reference patterns. The development of machine-learning-based tools for the analysis of µXRPD maps is subject of an ongoing PhD project, to further improve this time-consuming step.

### 2.5. Assessment of Radiation Damage

The risk of radiation damage must not be neglected, even more so in the context of the higher flux offered by EBS and for the study of hybrid and humid materials such as paint samples. Various actions have been initiated to assess, understand and mitigate radiation damage when analyzing cultural heritage materials [21,22]. Radiation damage manifests itself in at least two effects: (i) the modification of the sample, in particular its optical aspect but also its composition, which consequently leads to erroneous results in future chemical analyses; (ii) the alteration of the measured XRPD data (formation or disappearance of XRPD peaks, peak broadening). Radiation damage can be mitigated and prevented by reducing the X-ray dose hitting the sample, in particular by decreasing the flux and/or the dwell time [22]. In order to define the safety thresholds and establish safe procedures for the analyses of (historical) paint fragments in the context of the BAG, dedicated experiments were carried out on model samples to evaluate the two above points. Since paint samples constitute the majority of the materials under investigation within the BAG, radiation damage studies were carried out on a series of model oil paints prepared with different pigments. As an example, Figure 1 shows selected results obtained on a lead white pigment (mostly hydrocerussite). Some results obtained on chrome yellow pigments are presented as well in the Appendix A. To assess and understand the beam damage, optical microscopy images and Fourier-transform infrared (FTIR) spectro-microscopy maps were recorded at the ID21 beamline prior and posterior to a series of ID13 µXRPD maps, acquired at different flux, different dwell time and as single or triple acquisitions (see details in the Appendix A). XRPD patterns were also collected repeatedly at POIs, in different flux conditions, to monitor their evolution over time. The results show that with increasing flux and/or dwell time a yellowing of the sample is observed (Figure 1a). Furthermore, modification of the FTIR signal, notably in the C-H and C=O stretching mode domains, is seen, in particular with a decrease in C=O ester and the formation of C=O acid peaks (Figure 1b,c). After repeated XRPD acquisitions at the same point, a shift in position of the peaks, a decrease in the total diffracted intensity and a broadening of the diffraction peaks are observed. These changes are associated with amorphization leading to a loss of crystalline order in the material, thus reducing the intensity of the Bragg diffraction (Appendix A). Additionally, new peaks appear and can be partially attributed to the formation of metallic lead, Pb(0), through the reduction of Pb(II) (Figure 1d). Nevertheless, employing low flux (flux ~10^11^ ph/s), and short dwell time (10 ms), which were the conditions preferred for the analyses of painting fragments, the yellowing of the paint is not noticeable, and no modification is detected in the FTIR nor XRPD data (map 1 in Figure 1a–c, and scan 1 in Figure 1d) guaranteeing the validity of the measurement. The complete set of results of this study will be detailed in a forthcoming publication.

## 3. High-Angular Resolution X-ray Diffraction at ID22

### 3.1. Beamline Description

Pioneering studies using HR-XRPD for the study of cultural heritage materials were performed at the former BM16 beamline at the end of the 1990s [23], paving the way for the exploitation of synchrotron radiation in Heritage Science. In the field of cultural heritage, HR-XRPD is used for phase identification and quantification, microstructure characterization and, in some cases, complete structure determination [24,25]. Fast screening of a series of samples can also be carried out, a convenient way to look for example at modern reproductions obtained in the laboratory in controlled atmosphere (e.g., humidity, temperature, for aging/long-term degradation studies).

The ID22 beamline (formerly BM16 and ID31) combines a continuous range of incident energies (from 6 to 80 keV) with high brightness, also enhanced by the new EBS source, thus offering the possibility to carry out high-quality HR-XRPD. A highly monochromatic beam (∆E/E~10^−4^, E being the energy of the incident X-ray beam) of about ~1 × 1 mm^2^ and of low-divergence arrives on the sample, and diffracted photons are measured by scanning the 2θ circle which holds an EIGER2 2M-W CdTe pixel detector positioned behind a set of 13 Si(111) analyzer crystals. Because of the small acceptance of an analyzer crystal, precise 2θ angles of diffraction are defined, yielding very narrow resolution function, with a resulting FWHM of about 0.0025° (2θ) at 35 keV for the 111 reflection of a NIST Si 640c standard. The efficiency of detecting the diffracted radiation can be increased by operating multiple crystals in parallel [26], and thirteen channels, 2° apart from each other, are currently available. The presence of a 2D detector combined with the analyzer crystals offers additional flexibility in terms of data handling and processing [27], improving both peak shape at low diffraction angles and counting statistics at high diffraction angles, resulting in an overall increase in the quality of high-resolution powder diffraction data [28]. In the context of the BAG, standardized operating conditions are provided, with an incident radiation of 35 keV (λ = 0.3542 Å) to reduce absorption by the sample as well as limiting potential radiation damage. One of the requirements for the BAG is to have the samples compatible with the robotic sample changer (see part 3.2) in order to maximize data collection efficiency. However, more diverse setups (energy, sample stage, sample environment, etc.) are accessible through standard proposal calls.

### 3.2. Sample Preparation and Mounting

Two main types of samples are being analyzed through the BAG at ID22: free powders and artwork micro-fragments, both compatible with the use of borosilicate glass capillaries as sample holders. The choice of the capillary size for the free powders depends on the sample content and corresponding X-ray absorption coefficient, which can be calculated from https://11bm.xray.aps.anl.gov/absorb/absorb.php (accessed on 1 February 2022). In order to maximize the diffraction signal, a μR value (μ: linear absorption coefficient; R: capillary radius) below 1.5 is recommended. Capillaries are then mounted on a magnetic base, to be handled by the robotic sample changer in an automated way (see the robot at former ID31 in action: https://www.youtube.com/watch?v=ACMScnxOYkM (accessed on 1 February 2022). The most precious samples or samples requiring careful positioning are usually mounted manually on the spinner of the ID22 diffractometer.

### 3.3. Data Acquisition and Data Analysis

Similar to ID13, ID22 operates with the new BLISS control system and uses similar ICAT capabilities. In order to reduce preferred orientation, the samples are usually rotated (nominal speed close to 1000 rpm), and diffraction data are collected through continuous scanning of the 2θ circle at a chosen speed (from 0.5 to 30°/min, 2°/min being the default one) and at the appropriate acquisition time in order to achieve a step size of about 0.0005° (2θ). Several scans per sample can be collected to improve the statistics. The diffraction signals collected over the 13 channels (and over the different scans) are then combined using dedicated beamline software [29], and the final high-resolution powder diffraction pattern is obtained as a three-column ascii file (.xye). A new data processing suite taking full advantage of the EIGER2 detector behind the analyzer crystals is under implementation [28]. Data analysis can be carried out with various software offering crystal phase identification, microstructure analysis through Le Bail or Pawley fits, and phase quantification or structure studies through Rietveld refinement.

### 3.4. Assessment of Radiation Damage

The effects of radiation damage on the sample are sometimes visual, with a change of color (usually a darkening) under the X-ray beam. As observed during ID13 experiments, radiation damage usually results in diffraction peak shifting and/or broadening, and can be assessed by looking at the evolution of the diffraction signal between two scans. To reduce such effects, the incident beam can be attenuated, higher energies used, and faster scans implemented. If repeated scans are recorded on the same position, after comparison, the scans not showing signs of radiation damage can be merged to increase counting statistics. In the case of powders in capillaries, a fast scan is generally carried out (>15°/min), and repeated several times on fresh zones of the capillary [30].

## 4. Some Recent Examples of Studies Performed within the BAG Project

### 4.1. Revisiting the Bamyian Buddhist Paintings to Obtain More Insight into Pigment Syntheses and Early Oil Painting Practices in the Silk Road

As a first illustration of the BAG capabilities, Figure 2 reports an updated analysis of an iconic example, presented in a pioneer publication about the application of SR micro-analytical techniques for the analysis of painting fragments [31]. These fragments were sampled from 6th–9th C. Buddhist wall paintings from Bamiyan, Afghanistan, and analyzed at the ESRF in 2006. Thanks to SR-based micro-infrared analyses, the most surprising scientific outcome was the identification of the use of the oil technique in these very early wall paintings; nevertheless, results obtained with µXRPD/µXRF were also very informative. In particular, different forms of lead white were identified based on the variable cerussite to hydrocerussite ratios, revealing the use of different lead white qualities, or the different use (and evolution) of the same pigment. Information about the nature of degradation compounds (e.g., palmierite ((K,Na)_2_Pb(SO_4_)_2_), anglesite (PbSO_4_), moolooite (CuC_2_O_4_.nH_2_O) and atacamite (Cu_2_Cl(OH)_3_)) were obtained as well. At the time of these first experiments, the beam size available at the former ESRF ID18F beamline was 15 × 1 µm^2^ (hor.×ver.). The flux and detector technology imposed long acquisitions (5 s) but above all, a long lead time (10 s per pixel). To cover a corpus of ~40 samples, a compromise was made between map size and resolution, and maps were collected as 2 or 3 vertical profiles only, to preserve the resolution in the direction of the paint stratigraphy (vertical axis). Each map, of only ~2×100 pixels^2^ (hor.×ver.) necessitated about 50 min. With the present EBS-ID13 instrument, the same samples could be reanalyzed, with larger fields of view (up to 500 × 500 pixels^2^) covering the entire cross-sections, with a square and smaller pixel of 1 × 1 µm^2^, and still within a reasonable amount of time (only 18 min for the 500 × 500 pixels^2^ map). The larger field-of-view offers better statistics on the results. The smaller pixel size allows imaging smaller details. As an example, Figure 2 shows a comparison of data obtained in 2006 and in 2021, on thin sections from the same sample from Foladi cave 4, ca. 7–8th C. AD. The sample is a multi-layer system composed of (from depth to surface): a brown earthen plaster render covered with sizing layers, a white ground layer, a red layer, covered with a thin white layer. The final thick green layer highlights some whitish degraded aspect on its topmost surface. Ancient painters used such superimposition of complementary colors (here orange/white/green) to produce deep greens. XRPD data revealed that the two white layers are composed of different lead whites. The large and high resolution map acquired at ID13 reveals big cerussite particles (up to 50 µm diameter) in the ground white layer, while it appears much more finely ground (and with a higher amount of hydrocerussite) in the white layer above. This confirms a deliberate use of two distinct lead white qualities. Regarding the degradation layer, the ID13 map reveals that palmierite, anglesite, and atacamite form a quasi-continuous layer on top of the green layer. These new results provide a better understanding of the paint technologies used in these very early oil paintings and for their conservation.

### 4.2. Composition and Stability of Pigments Invented during the Industrialization Period (End of 18th- Beginning of 20th C.)

From the end of the 18th C., with the rise of modern chemistry and the discovery of new elements and new minerals, artists and craftsmen had new materials at their disposal, in particular pigments outperforming traditional pigments in tints and hues. This led to major evolutions in artistic practices, in paintings (e.g., impressionism) but also in glasses and ceramics. The more controlled synthesis procedure has an effect not only on the optical properties, but also on the morphology of the pigment, on its chemical composition and crystalline structure. Often, slight variations in the composition translate into a high variation of color, but also sometimes stability. HR-XRPD is therefore essential for distinguishing subtle variations in composition of crystalline artistic materials.

#### 4.2.1. Deepening the Knowledge of Formulations of Cadmium Red Pigments

Cadmium reds are a class of 20th century artists’ pigments described by the formula CdS_1−x_Se_x_. For their vivid orangish/reddish tone and excellent covering power, many well-known modern and contemporary painters, such as Jackson Pollock, often employed cadmium reds [32,33,34]. Recently, the extensive study of artificially aged oil paint mock-ups made up of CdS_1−x_Se_x_ with different x values, provided first evidence of the tendency of cadmium reds toward photo-degradation and proved that the conversion of CdS_1−x_Se_x_ to cadmium sulfates and/or oxalates is influenced by the oil binding medium and moisture and depends on the Se content [33]. Thus, the proper understanding of the overall pigment formulation and the stoichiometry of CdS_1−x_Se_x_ is highly relevant in view of the preventive conservation of paintings containing cadmium reds.

CdS_1−x_Se_x_ pigments have been largely studied by XRPD techniques [32,33]. In particular, the peak positions of the XRPD pattern are strongly affected by the x value, since the progressive substitution of an S atom with a larger Se one leads to a linear increase in unit cell parameters [35]. Furthermore, the accurate fit of the peak profiles can reveal microstructural features (such as crystallite size, and lattice defects and distortion) and provide quantitative information on both different crystalline phases and the amorphous phase. Thanks to the recent advancements of instrumental techniques and computational methods for data treatment it is often possible to obtain accurate structural and microstructural properties also from conventional X-ray sources, although the use of synchrotron radiation X-ray beams allows overcoming the intrinsic limitations of laboratory diffractometers, especially in terms of intensity, resolution and peak profile description.

As an example, Figure 3 shows a comparison of a portion of the Rietveld plots of an historical cadmium red pigment powder dated back to ca. 1960–1970 and produced by Kremer (hereafter called powder 442), in which the experimental XRPD pattern is compared with that calculated from a structural model, thus enabling the refinement of the above cited structural parameters (the Rietveld method) [36]. Figure 3a shows the (100), (002), and (101) peaks of CdS_1−x_Se_x_, collected with a laboratory diffractometer (PANalytical X’Pert Pro in reflection geometry, X’Celerator detector, Ni filtered CuK_α_ radiation). These peaks seem to show only two contributions, which come from two CdS_1−x_Se_x_ phases (indicated with I and II Roman numerals). The Rietveld refinement procedure successfully converged with reasonably low agreement factors, and the refined x values for the two phases are reported inside the box Figure 3a. Figure 3b shows the same portion collected at the ESRF beamline ID22 (35 keV radiation). With this pattern (it was possible to clearly identify five CdS_1−x_Se_x_ phases, indicated with I–V Roman numerals, some of them with a very small difference in Se content (refined x values are reported in Figure 3b). Figure 3c shows the detail of the (101) peaks of the ID22 pattern, in which the contribution of the five phases to the global profile has been reported in different colors. The weight % of the five phases plus barite are also reported, as obtained by the Rietveld analysis.

Note the fact that the data recorded with the synchrotron radiation X-ray source (Figure 3b) show a negligible instrumental contribution to the peak profile as compared to that recorded with the conventional X-ray source (Figure 3a). To obtain an idea of the instrumental contribution to the total peak broadening, it is sufficient to observe the sharpness of the 111 reflection of the added crystalline silicon as internal standard (marked with an asterisk in Figure 3b,c), as compared to the broadening of the other peaks. Being a highly crystalline sample with ideal crystallite size, the Si peak broadening can be specifically associated with instrumental effects. The virtual absence of instrumental broadening enables a better evaluation of the microstructural characteristics of the sample, which contribute to the diffraction peak shape. So, while the laboratory pattern might be reasonably well refined using an isotropic size and microstrain model, the peak shape of the ID22 pattern allowed the refinement of an anisotropic model, from which the average crystallite size and microstrain along different crystallographic axes can be estimated.

#### 4.2.2. Understanding Paint Degradation in Picasso Cadmium Yellows

The degradation of modern cadmium yellow paints (CdS/Cd_1−x_Zn_x_S) has been the object of intensive research [38,39,40,41,42,43]. The study of micro-fragments of historical paintings and of paint mock-ups established that the degradation process consists of the photo-oxidation of the original form of cadmium sulfide into cadmium sulfate. This process is fostered by environmental conditions [42] and by the presence of chlorine residues [40,41,43]. Research is still ongoing to assess how the degradation rate is influenced by the synthesis method, and in particular by the resulting pigment properties (e.g., particle size) and composition (presence of residues/secondary products) [44]. Further research on historical samples and paint mock-ups is still required to expand our knowledge on other factors influencing the degradation pathways of cadmium yellow paints.

Pablo Picasso’s *Femme* (*Époque des “Demoiselles d’Avignon”*, 1907, Fondation Beyeler, Riehen/Basel, Switzerland, Inv. 65.2) was the object of a wide conservation project, concerning the history and pictorial techniques of the painting. During this study, a comparison of the painting with an old slide from the museum archives revealed that the CdS-based paints had retained their original bright yellow color only in some areas, while they had turned brownish in other areas of the painting [45]. As these paints were subjected to the same environmental conditions and hence to the same natural ageing, they represent an important example for understanding the reasons behind the different stability of the various CdS-based paints. To evaluate the differences down to the micrometric level, two paint fragments from the different CdS areas were selected and studied employing SR µXRPD techniques (Figure 4).

Analysis conducted at ID13 confirmed that Picasso employed (at least) two different cadmium yellow paints [46]. The well-preserved yellow is a mixture of crystalline CdS (hexagonal and cubic forms) with two different extenders, lead white (hydrocerussite) and barium sulfate. In contrast, the now-brownish yellow is composed of amorphous or poorly crystalline CdS, indeed no diffraction peaks are present, but CdS was detected through µXANES measurements at S K-edge and Cd L_3_-edge at ID21. In this case, CdS is mixed only with one extender, barium sulfate. Additional crystalline compounds, such as cadmium hydroxychlorides, sulfates and carbonates, were also identified in the now-brownish layer, which can be associated either to residuals of the pigment synthesis method (hydroxychlorides and carbonates) [40,41,43] or to paint degradation (sulfates) [38,42]. This finding provides the first clear evidence of the different crystallinity of CdS in the two yellow paints employed by Picasso, a difference that can be ascribed to the production methods of the pigment [43,47]. The different crystallinity of CdS in the two paints, along with some residual of the starting reagents (i.e., Cd(OH)Cl), may have influenced the paint stability, leading to a severe degradation in paint layers where the poorly crystalline and highly reactive pigment with high presence of residues of the synthesis was used [48]. This finding paves the way to further tailored research on model paints in the framework of the Heritage BAG to correlate paint degradation with synthesis methods.

#### 4.2.3. Tracking the Origin of the Color of “Thénard’s Blue”, from the Manufacture Nationale de Sèvres

“Thénard’s Blue“ (CoAl_2_O_4_) is a spinel pigment, created in the early 19th century for the *Manufacture Nationale de Sèvres*. Its adaptation into a porcelain glaze required nearly 80 years of trial-and-error research. Experiments at ID13 and ID22 were combined to follow and understand, from structural and optical points of view, its complex evolution during the formation of a low-fire porcelain glaze [49]. To this aim, a series of pigments with composition Co_1−x_Al_2+2x/3_O_4_ (0 ≤ x ≤ 1) were prepared by co-precipitation and fired at 850–1400 °C. Then, porcelain glazes were prepared by mixing these pigments (33 w%) with a Pb-rich flux and firing at 880 °C (i) in a crucible to obtain powder samples and (ii) on a porcelain substrate (Figure 5a).

HR-XRPD at ID22 on powder samples was used to determine the crystal structure of the pigments, (i) alone, and (ii) embedded in the glaze after the firing process. Quantitative phase analysis with an internal standard enabled the determination of the solubility of the pigment and the composition of the glass phase, in particular the proportion of metallic oxides (Figure 5e). Complementary UV-visible spectroscopy showed the oxidation state of cobalt and its environment, and how its color was changed once in the glaze.

Besides, cross-sections of the glazed porcelain samples were observed by scanning electron microscopy coupled with an Energy Dispersive Spectrometer (SEM-EDS) analysis, and thin sections were analyzed at ID13 to acquire µXRF and µXRPD maps. SEM revealed the distribution of the pigment particles in the glaze (Figure 5c). EDS and µXRF detected Co, but could not differentiate Co from the glaze matrix and from the pigment due to the small size of the crystallites and the high penetration of electrons and X-rays, respectively. Conversely, the µXRPD maps allowed a selective analysis of the cobalt contained in the pigment, revealing the exact location of the spinel phase CoAl_2_O_4_. Comparing µXRF and µXRPD maps allowed understanding how cobalt dissolves and diffuses inside the glass matrix. As shown in Figure 5d, the Co-XRF signal (in red) is in proportion to CoAl_2_O_4_ XRPD signal (in blue) and is much more intense towards the surface of the glaze, indicating a higher concentration of dissolved cobalt at the glass surface.

Determining (i) the location of the cobalt in the glass phase, (ii) the location of the pigments in the glaze, (iii) the evolution of the spinel structure during firing, (iv) the amount of dissolved pigment and (v) the evolution of the color (linked to the oxidation state of the cobalt and its environment), gave us a good understanding of the dissolution and recrystallization mechanisms and will be discussed in a forthcoming paper. Eventually, the long pending issues of coloration and instability met when adapting Thénard’s Blue into a porcelain glaze found their explanation by this original combination of µXRF/µXRPD mapping on ID13 and HR-XRPD at ID22.

### 4.3. Applications to Conservation Studies

SR-techniques are used not only to identify the original materials used by craftsmen and artists and their possible degradation products, but they can also provide information about chemical reactions involved in conservation treatments and characterization of the penetration depth of the conservation products and efficiency of these treatments. As an example, Ca K-edge 2D µXANES has been recently used at beamline ID21, in combination to µXRPD mapping to study the stratigraphic distribution of calcium-based consolidants applied in limestones [50]. Some experiments in the BAG have been similarly dedicated to conservation purposes.

#### 4.3.1. Revealing the Interactions of Inorganic Conservation Treatments with Mg-Containing Frescos

The structural analysis of new crystalline phases formed in painted plasters after inorganic-mineral treatments and the investigation of their distribution within the porous matrixes are in high demand and a challenging task, as high phase selectivity, sensibility to trace phases and micrometric spatial resolution are simultaneously required.

Several inorganic-mineral treatments are available and ammonium oxalate (AmOx, (NH_4_)_2_C_2_O_4_·H_2_O) is one of the most widely used for the conservation of calcium carbonate stone materials (both natural and artificial) [51,52,53].

Here, a series of studies by µXRPD at ID13 and µXRF at ID21 (see technical details in Appendix A) focused on the interactions of AmOx treatment applied to Mg-containing historical frescos with the following aims:Identify the new oxalate phases crystallized after the AmOx treatment in the presence of Mg-rich and Ca-rich regions of the fresco;Localize the different oxalate phases with respect to each other, as well as to explore their distribution in the different regions of the fresco stratigraphy.

As an example, Figure 6 summarizes some μXRF and μXRPD outcomes collected in correspondence to the fresco painting. The stratigraphy of the fresco (shown in the optical image of Figure 6a as (1) plaster, (2) *intonachino*, (3) ~10 μm portion of *intonachino* with iron-based pigments) is well distinguishable in the μXRF distribution maps of Ca, Mg and Fe (Figure 6b).

Two different classes of reaction products are formed after the AmOx treatment: magnesium oxalates (glushinskite) and calcium oxalates (whewellite and weddellite). Magnesium oxalate and calcium oxalates are crystallized in different regions of the sample, with glushinskite formed close to the surface and whewellite localized in the sub-surface portion of the fresco painting (Figure 6c,d). The localization of weddellite has been studied but it is not discussed here. No iron-oxalates have been detected.

The oxalate phases are formed within the *intonachino* as well as in the plaster substrate. The comparison of the μXRF map of calcium and the μXRPD map of calcite shows that: (i) in the portion of *intonachino* with iron-based pigments (layer 3), calcium ions are all ascribed to calcium oxalates phases (no calcite); (ii) below the Fe-containing portion, the calcium ions are ascribed to both calcium oxalates phases and calcite of the matrix (Figure 6c). These findings demonstrate that in the iron-based region of *intonachino* most of the original calcite is converted to calcium oxalate phases. It follows that the oxalate framework restores the microstructural cohesion of the fresco stratigraphy. In addition, the newly formed calcium oxalate phases have a low solubility even to acid environments, providing an advantageous acid resistance for painted carbonate substrates exposed to polluted urban regions. Results of the complete study will be reported in a forthcoming publication. Above all, the significant results obtained thanks to the ESRF-EBS and the high-spatial resolution 2D µXRPD–μXRF mapping at ID13 and ID21 open up new insights in the field of conservation treatments applied to painted plasters and historical painted materials.

#### 4.3.2. Assessing Structural Damage in Wood Vessels

Even though most SR-based X-ray analyses are dedicated to evaluating the composition of inorganic pigments and paintings, the same techniques can also be used to analyze various materials, including (bio)organic materials, such as wood, and to contribute to their preservation. For instance, Sorres X, a 14th century cabotage ship, is a unique example of the few medieval vessels preserved in the Mediterranean [54]. It was unearthed in 1990, desalted, cleaned, treated, as usual, with polyethylene glycol (PEG) to consolidate its wood, and since 2011, it has been kept in the Museu Marítim de Barcelona (Figure 7a). It currently shows some recurrent sulfur and iron-containing efflorescence, which may be related to more severe underlying problems threatening its integrity (Figure 7b–d). Sulfur species (H_2_S, HS^−^ and S^2−^), common in marine environments, can lead to the formation of more oxidized species (sulfates, etc.), which can cause a significant volume expansion of the wood. In turn, iron species may lead to the formation of iron sulfide and pyrite or act as catalyzers and contribute to the degradation of wood components and PEG polymers into small organic acids (cf. [55,56], and references within).

In this context, the objective of the study is to determine the identity, extent and distribution of sulfur, iron and other mineral species within the wood, in order to assess internal wood damage and influencing factors (such as the kind of wood, depth, proximity to old iron bolts). Analyses by µXRF and µXRPD at ID13 allowed us to discriminate among chemical species, based on their distinct diffraction patterns (Figure 7e), and a precise mapping of crystalline compounds with high spatial resolution in large maps from microtomed cross-section slices (Figure 7f), such as the ones in Figure 7g. Sulfur and iron-containing salts with different hydration degrees and oxides are present, together with PEG, in different holm oak, pine and elm wood samples, showing the complexity of redox processes occurring on and within the wood. In this particular sample, gypsum (CaSO_4_·2H_2_O, in red) appears only as spots, apparently without a definite distribution, whereas jarosite (KFe_3_(SO_4_)_2_(OH)_6_, in blue) and lower amounts of pyrite (FeS, see the diffraction pattern) tend to accumulate near the surface (right side of the map) and around collapsed cells. This is consistent with the distribution of iron and sulfur, but also, for example, potassium and calcium, determined by µXRF (results not shown). PEG (in green) has penetrated through the rays (long thin tubular structures—such as the one on the upper left corner of the map—used for the radial conduction of water, minerals and organic substances in the plant) and into the cells, preventing them from collapsing. These rays may not only contribute to PEG diffusion, but also migration of salt and degrading bacteria [57].

Further analysis will help ascertain whether these compounds are ubiquitous and/or homogeneously distributed in the vessel (similar to the Mary Rose ship [56]) or if they tend to accumulate near the surface, as in the Baltic ships Vasa, the Crown and Riksnyckeln [58], all of them extensively studied hulls.

## 5. Conclusions

As shown in the various examples above, the instruments offered at ID22 and ID13 are highly complementary and useful for the characterization of cultural heritage materials. HR-XRPD is very efficient for the precise and sensitive detection of crystalline phases, their identification, and the characterization of their microstructural and structural properties. As shown in the examples above, the low detection limit allows us to detect minor phases, and the high angular resolution allows us to differentiate phases (pigments) with slightly different structures/stoichiometry. Complementarily, µXRPD imaging provides unique insight into the stratigraphical distribution of these phases at the micrometer scale. Although most of the applications concern paintings, the same techniques can be applied to ceramics, wood, etc.

The Historical Materials BAG started in fall 2021, and two beamtime sessions have already been allocated, two days at ID22 and four days at ID13. Thanks to the optimization of sample mounting and data acquisition, and together with the improved capabilities of the EBS source and the beamlines, 74 samples + 3 references, and a record of 186 samples + 2 references were analyzed by 12 and 15 end-users in these two experiments, respectively (with both on-site and remote control). Noteworthy, these end-users represent almost as many individual scientific projects and institutes, demonstrating clearly the high potential of the BAG system. In total, seven nationalities are represented (six from European countries). Beyond an increased efficiency to collect data, the BAG demonstrated a high impact in informing and training a new user community, in particular PhD students, who represent more than half of the users. Additionally, it provides the possibility for non-expert users to join a collective effort and to obtain support in all steps of the analytical workflow (from beamtime allocation, sample preparation, data collection, to data analysis). This new ad hoc access model is therefore a remarkable step forward to make synchrotron-based XRPD analyses a standard method for the characterization and preventive conservation of our cultural heritage.

## Figures and Tables

**Figure 1 molecules-27-01997-f001:**
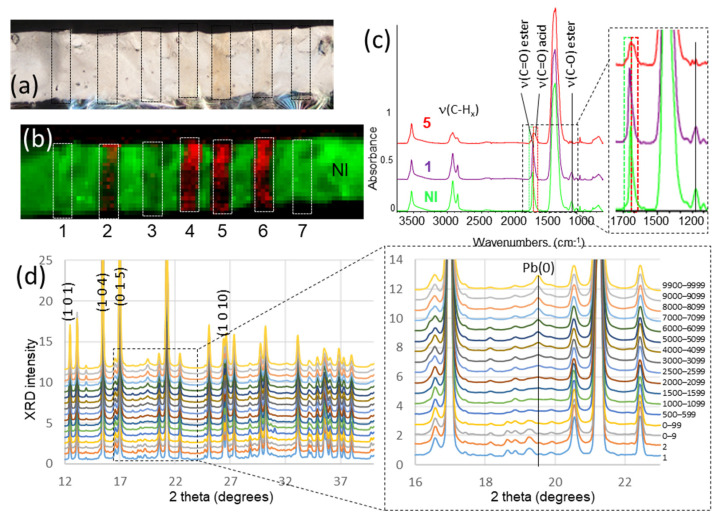
Assessment of radiation damage on a lead-white (mostly hydrocerussite) oil paint mock-up. Seven µXRPD maps were acquired at ID13, with the following conditions (1 or 3 repeats; different dwell times; at low flux (LF) or high flux (HF)): 1: 1 scan, 0.01 s, LF; 2: 1 scan, 0.01 s, HF; 3: 1 scan, 0.03 s, LF; 4: 1 scan, 0.03 s, HF; 5: 1 scan, 0.1 s, HF; 6: 3 scans, 0.01 s for each scan, HF; 7: 3 scans, 0.01 s for each scan, LF. The pixel size was 2.5 × 2.5 µm^2^ to avoid overlap between two consecutive points. The map width was 50 µm and the height sufficient to cover the entire thickness of the paint sample. The maps are represented as rectangles in (**a**,**b**). (**a**) Optical microscopy after µXRPD. (**b**) µFTIR map acquired in transmission mode (beam size 15 × 15 µm^2^, pixel size 10 × 10 µm^2^, 50 cumulated scans per spectrum) after performing the µXRPD maps. The red/green display shows the integrated intensity over the ν(CO) acid range (1683–1724 cm^−1^) and ν(CO) ester range (1726–1759 cm^−1^), respectively. These regions are displayed in (**c**) by a red and green rectangle, respectively. (**c**) Average FTIR spectra calculated over map 1, map 5 and a non-irradiated (NI) region. (**d**) XRPD 1D patterns measured with repeated acquisitions of 10 ms at LF; from bottom to top: 1st scan, 2nd scan, average of the first 10 scans, of the first 100 scans, and then average from x to x + 99, for x = 500, 1500, 2000, 2500, 3000, 4000, 5000, 6000, 7000, 8000, 9000, and 9900. The peaks assigned with Miller index (h k l) values are those for which the evolution of the position and full-width-at-half-maximum (FWHM) is reported in Appendix A.

**Figure 2 molecules-27-01997-f002:**
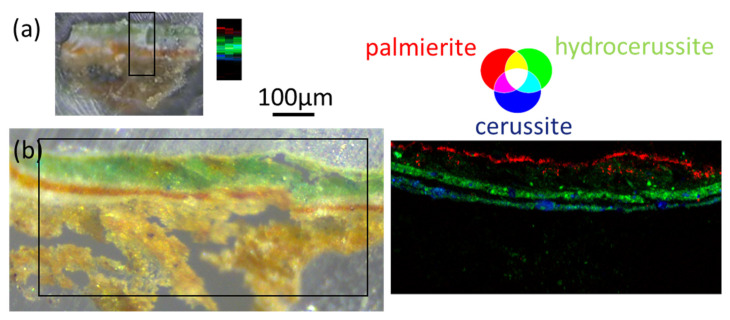
µXRPD maps of two sections of a fragment from a Bamiyan wall painting, acquired (**a**) in 2006 at the ESRF at the former ID18F beamline (map size: 150 × 60 µm^2^, pixel size: 1 × 20 µm^2^), and (**b**) in 2021 at beamline ID13 (map size: 800 × 370 µm^2^, pixel size: 1 × 1 µm^2^). The two acquisitions are displayed with the same scale (see text for technical details).

**Figure 3 molecules-27-01997-f003:**
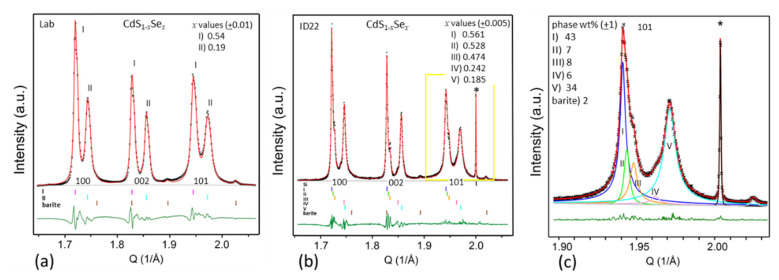
Portions of the Rietveld plots of the historical cadmium red pigment powder 442, showing the (100), (002), and (101) reflections of CdS_1−x_Se_x_ phases, collected using (**a**) the laboratory diffractometer and (**b**) the ESRF-ID22 beamline. (**c**) Enlargement of the frame marked in yellow in (**b**) showing, in different colors, the contributions of the five CdS_1−x_Se_x_ phases to the (101) reflection. The refined x values (and their standard deviation) for the different phases are reported in box (**a**) and (**b**), while the weight % of the five identified phases, plus barite, are reported in box (**c**), as resulting from the Rietveld analysis [37]. Black cross: experimental data; red line: calculated profile; dark green line: difference curve. Vertical marks indicate the calculated positions of Bragg reflections for each crystalline phase. The peak of silicon, used as internal standard, is marked with an asterisk (*).

**Figure 4 molecules-27-01997-f004:**
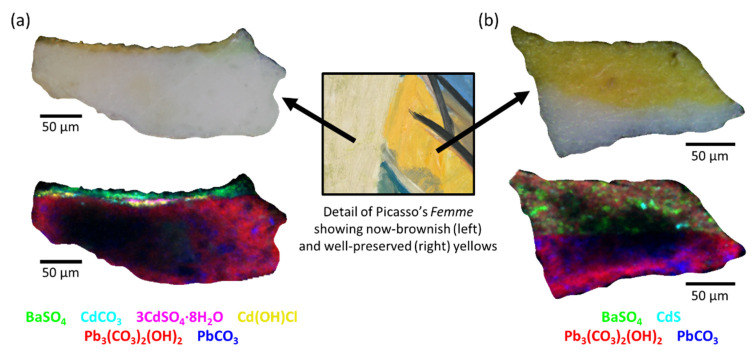
Detail of Picasso’s *Femme* containing both now-brownish (left) and vibrant (right) yellows. Visible image and SR-µXRPD distribution maps of crystalline phases identified in (**a**) now-brownish and (**b**) vibrant yellow micro-samples, extracted from representative areas of the painting. Samples were prepared as cross-sections embedded in resin. The colors of the different crystalline compounds identified are indicated in legend. The darker region in the center of the µXRPD maps is due to the poor transparency of the sample to X-rays (measurements acquired in transmission mode).

**Figure 5 molecules-27-01997-f005:**
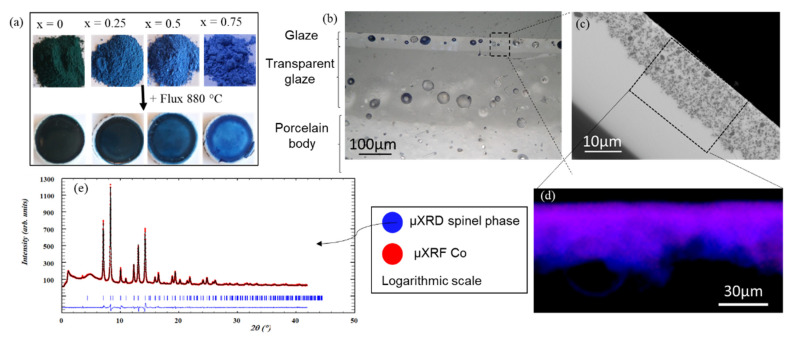
Study of the stability of Co_1−x_Al_2+2x/3_O_4_ spinel pigments in a porcelain glaze. (**a**) Pigments x = 0; 0.25; 0.5 and 0.75 fired at 1000 °C and their respective glazes; (**b**) Optical microscopy image of the cross-section of a painted porcelain; (**c**) SEM image of the glaze layer of glazed sample (x = 0.25 at 1000 °C); (**d**) µXRF and µXRPD map of glazed sample (x = 0.25 at 1000 °C) acquired on ID13, blue = µXRPD signal of the spinel phase, red = µXRF signal of Co (K-edge); the XRPD pattern of CoAl_2_O_4_ in the glaze is then compared with the (**e**) HR-XRPD diagram of the pigment sample, (x = 0.25, at 1000 °C) acquired on ID22 (35 keV).

**Figure 6 molecules-27-01997-f006:**
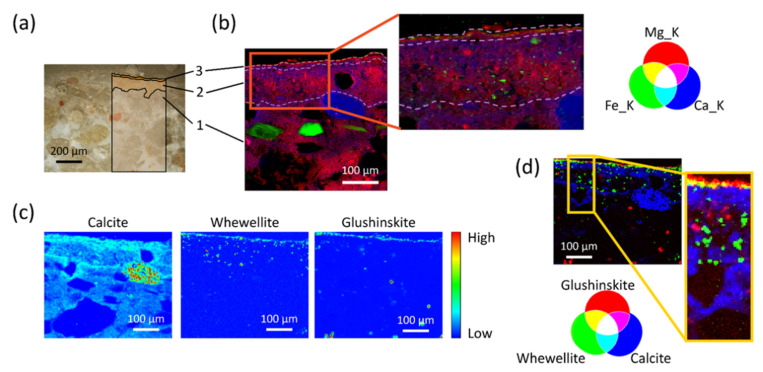
Analysis of the interactions of inorganic-mineral conservation treatments (AmOx) with Mg-containing frescos by a combination of μXRPD and μXRF mapping carried out at ID13 and at ID21, respectively. (**a**) Optical image of the fresco stratigraphy in cross-section: (1) plaster, (2) *intonachino*, (3) ~10 μm external portion of *intonachino* with iron-based pigments. The investigated ROI was about ~500 × 400 μm^2^ at ID13 and ~ 400 × 400 μm^2^ at ID21. (**b**) RGB correlation of μXRF distribution maps of calcium (Ca K), magnesium (Mg K) and iron (Fe K). The inset highlights the presence of iron-based pigments in the external portion of the *intonachino*; (**c**) μXRPD distribution maps of calcite, whewellite and glushinskite presented in a colourmap spanning from low (blue) to high (red) values of relative intensity; (**d**) RGB correlation of the μXRPD distribution maps of calcite, whewellite and glushinskite.

**Figure 7 molecules-27-01997-f007:**
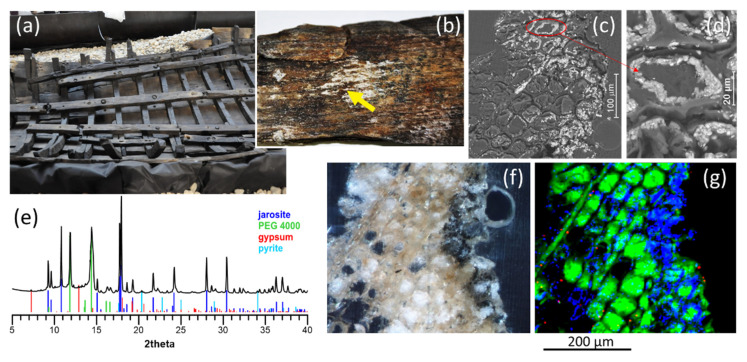
Study of the degradation of a waterlogged medieval timber hull. (**a**) View of the vessel Sorres X (Museu Marítim de Barcelona), which has a total length of 9.5–10 m. (**b**) Detail of efflorescence on pine wood. Back-scattering SEM images of (**c**) a thin cross section of the same pine wood sample and (**d**) detail showing crystal accumulations on the inner side of the cell wall. (**e**) Average diffractogram of a transversal 20 µm-microtomed section of the same pine wood sample, compared to reference patterns and (**f**) corresponding microscopic image and (**g**) False-color µXRPD map, where the inner part of the wood is on the left and the wood surface is on the right of the images. Colors in the µXRPD map represent PEG (green), jarosite (blue) and gypsum (red); 456 × 386 µm^2^ map, obtained with a 2.5 × 2.5 µm^2^ beam, with a step of 1 × 1 µm^2^.

## Data Availability

The data presented in this study are available on request from the corresponding author.

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
