# Peer review of "The “Historical Materials BAG”: A New Facilitated Access to Synchrotron X-ray Diffraction Analyses for Cultural Heritage Materials at the European Synchrotron Radiation Facility"

_molecules, 2022, doi:10.3390/molecules27061997_

Round 1
Reviewer 1 Report
The work entitled: "The “historical materials BAG”: a new facilitated access to synchrotron X-ray diffraction analyses for cultural heritage materials at the European Synchrotron Radiation Facility" by Marine Cotte et. al. is an excellent review article. As an ESRF user I have been appreciating the exceptional scientific standards and high quality of works related to ESRF. This particular paper is well-written clearly organized and will be an excellent advertisement on the work related to investigation of historic materials by means of synchrotron radiation at the ESRF .
Author Response
Answer: we warmly thank the reviewer for his/her comment.
Reviewer 2 Report
The manuscript by M. Cotte at al. “The “historical materials BAG”: a new facilitated access to synchrotron X-ray diffraction analyses for cultural heritage materials at the European Synchrotron Radiation Facility” presents the new experimental capabilities that are offered at ESRF after its upgrade and the boost of performance of beamlines ID13 and ID22. The specific hardware and software developments to facilitate and speed-up data acquisition and data processing are described in detail in the manuscript. The advantage of these new tools is illustrated by the examples of the studies of cultural heritage (CH) objects carried out within the framework of the “block allocation group ( BAG)” project. Introduced in BAG a new model of collective access to the beamtime combines in one project the users who need the same techniques. The scientists of 10 European institutes used this collective access to beamline to study the cultural heritage materials. The first outcomes of these studies applicated to pigments, paintings, ceramics and wood are described in the manuscript. The radiation techniques are used widely to characterize the structure of ancient materials. High- angular resolution X-ray powder diffraction is very efficient for the precise and sensitive detection of crystalline phases, their identifiсation and the characterization of their microstructural and structural properties. The structural characterization of historical materials help to understand the early life of the materials and later phases of it life (degradation, past and present conservation interventions). Determining precisely the pigment composition is very important for conservation purposes. Crystalline phase identification, determining the pigment composition and its distribution within a multi-layer system, reveal the specificity of a painting process and can highlight materials which were not in the original artwork, for example materials added later. In the manuscript similar investigations were carried out with the wellknown works of art (Bamyian Buddhist paintings, Pablo Picasso’ s pictures).The results obtained described in the manuscript shows that new tools of the ESRF together with new access model to beamtime could make synchrotron-based X-ray diffraction analyses by a standard method for the characterization and preventive conservation of our cultural heritage. This manuscript is a result of successful collaboration between the specialists in the natural sciences and the humanities and may be of interest for a wide range of readers. The manuscript can be published in the present form.
Author Response
We warmly thank the reviewer for his/her comment.
Reviewer 3 Report
Summary: In this paper, authors described the capabilities offered by the recently implemented X-ray diffraction machines with advanced mapping and resolution systems. In addition, they have shown the first outcomes of this new access. I think one of the uniqueness of this system is to provide ability to access different type of materials. Most XRDs are reserved for certain type of materials to prevent contamination. Another important property of this system is to allow remote access to facility by anyone worldwide. I should note that that sensitivity and precision of the machine is impressive.
Comment1: Can data be transferred to any other type of file than .h5 files? If so, could you please add this to the manuscript. I think it is important for worldwide users to be able to plot the data without having access to a specific software.
Comment 2: Are there any other radiation damage studies has been done on other colors than white? If so, can authors provide that data in the supplementary file. If not, could you please comment on how we can generalize the radiation damage effects on white colors to the other colors.
Comment 3: Authors claimed that radiation damage usually results in diffraction peak shifting and/or broadening. In figure 1 neither the peak shift nor the peak broadening was obvious. I would recommend authors to show the peak shift
Comment 4: I would recommend finding full width half maximum (FWHM) of the different lines and plot them or state it in the table. Figure 3-line broadening is not obvious. For example, you can plot FWHM vs X values.
Comment 5: What are the colorful lines i.e yellow, purple indicates in Figure 3C. If they are fitted values, could you please indicate so and let us know what did you use to fit these values. Seems like they are phase wt%. Could you please be clearer on that. Also, in the same images Figure 3B x values were cut. Could you please put another image where x-values were not cut?
Comment 6: How would you account for the electron irradiation affect through SEM images?
Comment 7: In Figure 5c values below the figure can’t be read. Also, the axis of figure 4e is hard to read. Could you please replot those images?
